# Data Reduction for Low Energy Nuclear Physics Experiments Using Data Frames

**Caleb A. Marshall**[1,2*]

**1** Institute For Nuclear & Particle Physics, Ohio University, Athens, OH, USA
**2** Facility For Rare Isotope Beams, East Lansing, MI, USA

* camarsha@unc.edu

## Abstract

Low energy nuclear physics experiments are transitioning towards fully digital data acquisition systems. Realizing the gains in flexibility afforded by these systems relies on equally flexible data reduction techniques. In this paper, methods utilizing data frames and in-memory techniques to work with data, including data from self-triggering, digital data acquisition systems, are discussed within the context of a Python package, `sauce`. It is shown that data frame operations can encompass common analysis needs and allow interactive data analysis. Two event building techniques, dubbed referenced and referenceless event building, are shown to provide a means to transform raw list mode data into correlated multi-detector events. These techniques are demonstrated in the analysis of two example data sets.

## 1 Introduction

Low energy nuclear physics is concerned with the structure, properties, and origin of nuclei. As a consequence of the broad reach of this field, complementary experimental efforts happen at both billon dollar facilities with collaborations of several hundred participants, to university run labs where critical experimental information might be the product a few individuals[1]. While the trend has been slow, both national user facilities and university labs have transitioned or are transitioning away from traditional analog electronics and towards fully digital data acquisition systems (DAQ)[2].These digital systems have removed the need for complex analog setups that handle triggering and signal processing. As a result, the data processing pipeline, both online and offline, is now tasked with software implementations of much of the logic that would have previously been handled by nuclear electronics, thus placing additional requirements on the field's data analysis tools.

Currently, a stumbling block for this transition to digital DAQs is that low energy nuclear physics frequently adopts the analysis tools and software of the high energy particle physics community, despite the significant structural differences between the data produced by the two fields. As an example of this difference, consider two of the most prominent experiments from the 1990s in both fields: the CDF and DØ experiments at Fermilab responsible for the discovery of the top quark in 1995 [1, 2] for the high energy particle physics community, and the construction and operation of Gammasphere at Lawrence Berkeley and Argonne National Labs in the low energy nuclear physics community. Both DØ and CDF produced events (time correlated collections of all detector signals) of around 200 KB in size [3]. Gammasphere, however, was producing events of only 100 B [4]. Events in low energy nuclear physics are more numerous and nearly all detector information can be recorded, yet they contain significantly less information (hundreds to thousands of channels with low detector multiplicity versus hundreds of thousands to millions of channels with high detector multiplicity). The final volume of the data sets from the two fields might be comparable, it is reasonable to say that the low energy nuclear physics data is distributed among many more experiments making individual data sets orders of magnitude smaller. As a result, analysis methods tailored to smaller events sizes and fewer channels greatly benefit the field, and computational resources can be leveraged to allow efficient data exploration to speed up analysis.

In this paper an analysis framework designed specifically for smaller scale low energy nuclear physics will be discussed. This framework utilizes data frames for in-memory analysis (all the data can fit into random access memory (RAM)) encouraging rapid and interactive data exploration. Although designed with digital DAQs in mind, many of the benefits of the framework can be utilized for analog systems as well. A concrete implementation of the framework's principles resides in a Python package called `sauce`[3], which will be used in examples throughout the paper. However it is the goal of this paper to discuss the recurring challenges that arise when dealing with digital DAQs and data reduction in as much generality as possible.

---

[1]https://aruna.physics.fsu.edu/

[2]This transition has been slowest among applications that had little need for the greater throughput of the digital systems, but would suffer from the degraded timing and spectroscopic information. The gap in energy and timing resolution between digital and analog systems has narrowed considerably over the last decade.

[3]https://github.com/camarsha/sauce

## 2  Motivation and Design Choices

For the remainder of this paper, code snippets will be given frequently. It is assumed the reader is familiar with the syntax of the Python programming language. Rapid data exploration is aided by minimizing the amount of code required to carry out common analysis tasks. Consider what it would take to be able to find the timing difference between two detectors in one line of code:

```
dt = det0.time - det1.time
```

Where this single line of code represents an array operation such that every timestamp recorded for det0 is subtracted from those of det1. For this operation to be meaningful, it is required that the two timestamp arrays be equal in length and ordered so that only the difference of *related* timestamps is computed. Ensuring these two conditions are met is difficult, and so it is far more common to write some variation of the following:

```
for i, event in enumerate(events):
    if det0.time and det1.time:
        dt[i] = det0.time - det1.time
```

Ignoring how a collection of events was even constructed to begin with, this small snippet of code already poses problems for exploratory analysis. First, we are forced to sort through every event to find the few that we are interested in, which can be computationally expensive. Second, any further analysis requires we either write many such loops, or worse that we add more logic within the body of the existing loop. Third, such granular logic is prone to introducing errors. This last point is especially important in the context of a low energy nuclear physics experiment, where many analysis will start directly from the list mode data of the DAQ. Experimenters will frequently be implementing their own sorting routines, making the lower level code of the later sample a larger liability for the correctness and efficiency of any subsequent analysis. The additional complexity that comes with digital DAQs further exacerbates these problems.

We can avoid these issues entirely by developing a scheme that will enable the simple declarative nature of the former code sample. Doing so will require we have effective ways of handling list mode data and event building, which, in turn, requires we find a suitable data structure to hold and work with these data. The argument will be made that data frames serve this purpose well in Sec. 2.2, but first a discussion on some of the pitfalls of other data structures is merited.

### 2.1  Representation of List Mode Data

Modeling our data as a collection of events that hold the information for each channel in the system closely mirrors the data flow from a traditional analog signal processing setup, and could be in part why such a data pipeline is so common. To clarify the operation of such a setup, a peak sensing analog to digital converter (ADC) is coupled to analog signal processing and logic. Prior to data recording, a trigger logic is decided upon and implemented. The ADC gate will open and record data when signals of interest are present. Offline analysis involves sifting through these "events", which consist of the pulse height information of all the channels that fired within the ADC gate. Correlations between detectors are completely dictated by the hardware trigger logic and cannot be altered in software. In this case, it is simple to represent the DAQ's list mode data as a single $N$-dimensional array of pulse heights as shown in Fig. 1. While attractively simple, this approach is memory inefficient. As the number of channels in the system grows, the $N$-dimensional array will increase in size as well, but critically its memory consumption is only tied to the number of channels and the system wide event rate. The same amount of memory is required for each event regardless of how many channels fired.

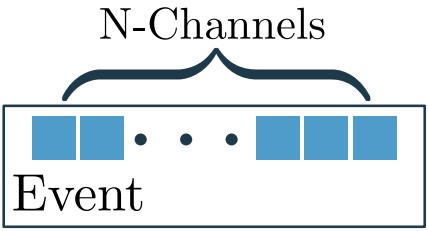

Figure 1: Simple event struture for a peak sensing ADC. The pulse height recorded in each channel is one element of an $N$-dimensional array.

As an example, consider a system with 128 channels (four 32-channel ADCs) with 16 bit energy resolution firing at a total rate of 10 kHz. If a fixed length array is chosen to represent these data, then 9 GB of data would be produced in one hour. The size of this data can be unwieldy to deal with even if it fits into memory, as a loop will have to sift over 36 million events for any operation on the data. It is likely to push an experimenter towards a workflow of a single monolithic event loop to populate histograms, viewing those histograms, and then iterating this cycle until the analysis is finalized.

Sticking to the analog case for the moment, what if we were to only process the data from the channels that fired in an event? We would need to identify each channel that fired with a channel number (conservatively assumed to be 8 bits in this case) and an event number to group the channel hits that belong to the same event (conservatively assumed to be 32 bits). Software logic would then be needed to reconstruct events. However, the amount of data produced in an hour is now dependent on the average number of channels that fire, ranging from 250 MB if one channel fires on average to 32 GB if all channels fire. Less data is produced on average than the $N$-dimensional array case for average channel multiplicities below 35. It also seems more reasonable to start an analysis working on individual channels and then looking at event information only when needed.

Now consider self-triggering digital DAQs, where each channel in the system triggers independently and records any data that passes an electronic threshold to disk. Pulse height information is now accompanied by timestamps and there is no longer a hardware definition of an "event" [5]. Instead, we must start our data reduction with raw list mode data from the DAQ, which for simplicity can be thought of as a tuple of numbers:

$$hit = (channel, adc, timestamp). \tag{1}$$

In this case, $channel$ is a general variable that uniquely identifies an electronic channel in the system, $adc$ is a digitized pulse height with arbitrary units, and $timestamp$ is an absolute digital timestamp with a resolution that is system dependent. The $N$-dimensional array is no longer an obvious or convenient choice for a data structure, since no channel in the system can be correlated with another until a decision has been made about how to build events in software. A lack of trigger logic also means that when events are built, multiple hits from a channel could be present within a single event. If we were to implement the simple $N$-dimensional array event model, we would now need jagged arrays (Fig. 2) where each element of an array can have any number of sub-elements [6,7]. Assuming 64 bit timestamps, the 9 GB of the analog case would now become 46 GB, and it is now even harder to work with the data in memory, which will force our event loop to stream data from disk, slowing it down even further. This case also forces us to adopt an event building scheme before even seeing the data,

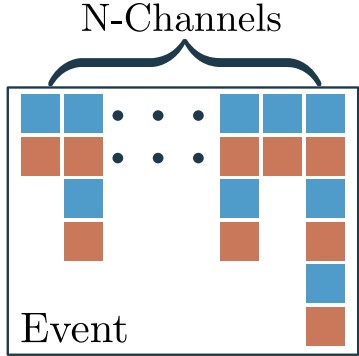

Figure 2: Simple event struture for a self-triggering DAQ. The pulse height (blue) is now recorded with a timestamp (red) in each channel. Each channel can fire any number of times within an event leading to an **N**-dimensional jagged array.

140 making it difficult to judge the effects of the event builder's parameters like the build window.
141 By trying to avoid the inherent complexity of a trigger-less digital DAQ and structuring the
142 data from them as a set of events, we are lead towards an imperative programming style and
143 a less interactive analysis.

## 2.2 Data Frames

145 It is now clear that effective analysis of data from a digital DAQ requires a data structure
146 that is suited towards working with the raw list mode data of Eq. (1). The chosen data struc-
147 tures should have a set of operations that closely map to typical analysis tasks, such as applying
148 thresholds or energy calibration, while also providing tools to avoid the intensive memory costs
149 of sparse jagged arrays. The choice made for `sauce` and that will be discussed for the rest of
150 the paper is to use data frames for these tasks. Briefly, data frames are a type of columnar data
151 structure that combine properties of relational tables and matrices [8]. They were first intro-
152 duced in the context of the S programming language [9] and have since become widespread
153 tools for data analysis. Implementations of data frames exist in numerous programming lan-
154 guages, including R [10] and Python [11, 12]. CERN's `ROOT`, a common choice for analysis
155 in low energy nuclear physics, also implements data frames as `RDataFrame` [13, 14]. The
156 version of `sauce` described in this paper has been implemented using the Python bindings of
157 the `polars`[4] library.
158     To clarify the structure of the `sauce`'s channel data frames, recall the basic list mode datum
159 given in Eq. 1. For each channel identifier, a separate data frame would be created such that we
160 have a columnar data set of *adc* and *timestamp* values. Fig. 3 shows the transformation of
161 list mode data into channel specific data frames which are named to uniquely identify them.
162 Notice that as a consequence of this storage scheme pulse heights are naturally associated
163 with their time stamps and visa-versa. If the DAQ produces additional information (e.g pile-up
164 detection or waveform data) it can simply become another column. In `sauce`, these structures
165 are implemented as a class called `Detector`, which stores a data frame along with an identifier
166 (name). There is no requirement that data frames for individual channels be the same length
167 or have the same number of columns. When it becomes necessary for the analysis to compare
168 hits in different channels, multi-channel data frames will be constructed programmatically
169 using event building (Sec. 3).

---

[4]https://pola.rs/

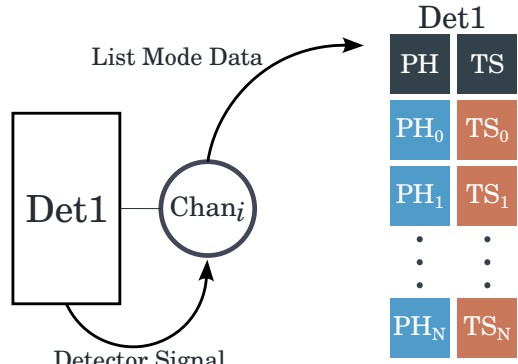

Figure 3: Sketch of the transformation of list mode data to data frames that occurs for each channel. `sauce` associates each data frame with a name in a class called `Detector` so that it can be identified when combining data from multiple channels. PH and TS stand for pulse height and timestamp, respectively.

## 2.3 Basic Nuclear Physics Analysis as Data Frame Operations

There has been some effort in to formalize data frame operations in Ref. [8] (see Table 1 in that work), which are partially derived from Ref. [15]. However, in this discussion I will present code snippets showing the necessary `polars` operations to keep this work grounded in their practical use. While this removes some generality from the discussion, it has the benefit of showing the utility of data frames for this application, and how little code is necessary to carry out tasks. Many `polars` operations are analogous to similarly named SQL functions or share names with common higher-order functions from the functional programming paradigm (`filter`, `reduce`, etc.).

Before moving on to `sauce`, let's look at some basic analysis using data frame operations directly. A hypothetical data set comprised of all hits in a single data taking run has three columns: channel, adc, time. The data set has been saved as a parquet file. Loading the entire data set into memory is done with:

```
run = pl.read_parquet("run_data.parquet")
```

Our simplified experimental setup system of only two detectors occupying channels 0 and 1 of the system, respectively, and dubbed, affectionately, "det0" and "det1". The hits belonging to "det0" can be found by selecting the rows that have the number 0 in the "channel" column and likewise for "det1":

```
det0 = run.filter(pl.col("channel") == 0)
det1 = run.filter(pl.col("channel") == 1)
```

Now `det0` and `det1` are data frames that only consist of the hits that occurred in those channels of the system. As a result, if we wanted to look at the pulse height spectrum, we would merely need to select the "adc" column and then histogram it (for example using the `histogram` function of numpy [16]). Column selection can be done with either:

```
det0.select(pl.col("adc"))
```

or

```
det0["adc"]
```

Energy calibration can be taken care of in one line of code per detector. Assuming a linear calibration with slope $a$ and intercept $b$ that will give us units of keV:

```
det0 = det0.with_columns(
    (a * pl.col("adc") + b).alias("energy")
)
```

A new column, "energy", is added to the data frame that has the energy calibrated values. Now an energy threshold can be applied to only keep hits above **100** keV. Again using the filter operation:

```
det0 = det0.filter(pl.col("energy") > 100.0)
```

The toy problems above show that data frames give us tools to act on entire columns without having to write loops. However, these are extremely simplified examples. Any real analysis would quickly run into bookkeeping issues when the number of detectors is large, and the timing information is useless until events can be built to compare timing differences between detectors. As mentioned at the end of Sec. 2.2, sauce wraps data frames in a `Detector` class that associates data frames with named identifiers. Methods for these classes are also implemented to make the expressions from above less verbose and to ensure in-place (destructive) modification of data frames to cut down on unnecessary copying. Utilizing this class, the above becomes for "det0":

```
det0 = sauce.Detector("det0")
det0.find_hits("run_data.parquet", channel=0)
det0["energy"] = a * det0["PH"] + b
det0.apply_threshold(100.0, axis="energy")
```

where it is understood that the parameter `axis` is sauce's terminology for a column name. The `polars` data can still be accessed directly using:

```
# get the data frame contained in the detector.
det0.data
```

which can be necessary for more complex analysis (see Sec. 4.2).

Before moving on to event building, two additional operations need to be defined that do not have the same immediate utility of the operations above, but that will be essential for dealing with the complexities of real detectors. The first is the union or concatenation of data frames. By applying a union to data frames from several channels, a single aggregate data frame will be returned that includes the hits of all the input channels. This is a useful operation for physical detectors that are readout with many channels. In sauce we write:

```
# returns a new detector called det_0&1
sauce.detector_union("det_0&1", det0, det1)
```

However, once a union has been performed, it is no longer possible to track which channel produced a hit. In order to recover this information, a "tagging" operation has been introduced. A tag merely adds a constant value to every hit in a data frame. If a `Detector` is tagged prior to a union, the resulting "tag" column will allow us to recover the individual channel information.

```
# create a new column called "tag" filled with 0
det0.tag(0)
```

To see these two operations used in practice refer to Sec. 4.2, which treats a position sensitive micro-channel plate detector (MCP) with four corners as a single detector with a tag "corner".

# 3  Event Building

A self-triggering DAQ cannot group channel data together without the concept of event building. Hits in two channels can only be said to be related if their timestamps fall within a specified time interval, ultimately based on both physical (time-of-flight, lifetimes, etc.) and electronic properties (signal delay, overhead in signal processing, etc.). Hits that fall within the time interval can be assigned an event number, and at later stage in the analysis these

event numbers can be used to examine correlations between the separate channels by combining their data frames. The basic operations covered in the last section are of little use if we cannot relate the hits in one channel to another. However, once events are built, nearly any analysis task can be defined using some combination of basic data frame operations and event building. Below, two schemes for event building are described that cover a majority of common use cases. They are dubbed *referenceless* and *referenced* event building. The former allows any channel hit to start an event building period, and is well suited to working with a single detector that has multiple segments being readout into separate channels (e.g. a HPGe clover detector or segmented silicon detector). The latter is more suited for looking at coincidences from distinct detectors, since it will use only selected channels to build events (e.g. particle-gamma coincidence measurement of Sec.4.1).

Before going further, all the data frames for the channels are assumed to be sorted by their timestamps such that they are increasing:

$$t_0 \leq t_1 \leq t_2 \leq \ldots, \tag{2}$$

this is a trivial procedure to carry out with data frames, and the `Detector` class does it automatically when the data is loaded and when operations that could alter the order are carried out (e.g. a union operation).

## 3.1 Referenceless Event Building

Given that we have a set of $n$ channel hits that are time ordered from $t_0$ to $t_{n-1}$, a referenceless event builder can be defined using only a single parameter, the build window $\Delta t$. Starting from the earliest hit, $t_0$, and beginning to enumerate the events, $evt = 0$, all subsequent hits, $t_i$ that satisfy $t_0 \leq t_i \leq t_0 + \Delta t$ are assigned $evt = 0$. The next hit that does not satisfy the condition, call it $t_m$, is taken as the start of the next build window, $evt$ is incremented, and all hits within $t_m \leq t_i < t_m + \Delta t$ are assigned $evt = 1$. The steps are repeated until all hits have been assigned to an event. Event building of this type is the equivalent procedure in analog electronics of using a logical OR from the channels to open an ADC gate. A sketch of this procedure is shown in Fig. 5, and pseudo-code is given in Fig. 4.

It is possible with this technique to lose true coincidences due to background counts opening a build window which closes before all of the coincident hits are processed. When this happens, true coincident hits will be assigned different event numbers and effectively be lost. By considering two Poisson processes that generate the background counts and true coincidences, a rough estimate can be made to quantify the impact of the event builder failing to properly group hits. Let $t_r$, $\lambda_r$, $t_n$, and $\lambda_n$ be the timestamps (in seconds) and rates of a hit of interest and noise (in Hz), respectively. Assuming coincident hits always come at a fixed time from $t_r$, we define a coincident time interval $\delta t_{coin}$. The number of dropped hits is the expected number of background counts occurring at a time such that $(t_r - t_n) + \delta t_{coin} > \Delta t$. Since $t_r$ and $t_n$ are independent, the chances of this happening is simply the expected number of noise counts in the time interval $\delta t_{coin}$. Expressing the ratio of the number of measured coincidences after event building, $N_{meas}$, to the true number of coincidences, $N_{true}$, gives:

$$\frac{N_{meas}}{N_{true}} = \delta t_{coin} \lambda_n. \tag{3}$$

Note that this is independent of the build window and only depends on the timing difference between the signal of interest and noise rate. A Monte-Carlo simulation was carried out that simulated the two process explicitly, and was found to agree with Eq. 3 for noise rates below **1** MHz and build windows within a factor of **2** of $\delta t_{coin}$. For measurements that have high backgrounds relative to the signal of interest, there is a risk of a referenceless event builder

```python
def build_referenceless_events(time_array, build_window):

    t_i = time_array[0]  # start of window
    t_f = t_i + build_window  # end of window
    event = 0
    # array that will hold the event numbers
    event_numbers = np.empty(len(time_array))

    for i in range(len(time_array)):
        # current timestamp
        tc = time_array[i]
        # if it is within the window, assign it an event number
        if tc >= t_i and tc < t_f:
            event_numbers[i] = event
        # else it is its own event, and starts a new window
        else:
            t_i = tc
            t_f = tc + build_window
            event += 1
            event_numbers[i] = event
    return event_numbers
```

Figure 4: Referenceless event builder in `Python` using `numpy` arrays.

290 dropping a significant fraction of coincidences due to high rate channels; however, by using a
291 digital DAQ timestamps can be shifted such that $\delta t_{coin}$ can be made arbitrarily small, down
292 to the timing resolution of the system. If this precaution is taken, referenceless event building
293 is a viable choice for a system wide event building scheme. However, practically, referenceless
294 event building on a system wide scale leads to complex data reduction. Since it does not en-
295 force a one-to-one relationship between hits in different channels, additional logic needs to be
296 implemented by the experimenter in order to examine coincidences. Due to these considera-
297 tions, referenceless event building is included in `sauce` as a method of the `Detector` class.
298 An example:

```python
# build referenceless events
det0.build_referenceless_events(500.0)
det0["event_det0"] # event numbers
det0["multiplicity"] # number of hits in an event
```

303 The number passed to the event builder is the $\Delta t$ (i.e the build window) from above and uses
304 the units of the timestamp column. A "multiplicity" column is added automatically, since it is
305 frequently needed as a diagnostic after referenced event building.

## 3.2 Referenced Event Building

307 The referenced event building procedure differs from the referenceless builder discussed above
308 because it gives priority to selected channels. Reference channels are selected from the system
309 and their timestamps are used to build a set of non-overlapping time windows. If some number
310 of windows would overlap, then only the window defined by the earliest timestamp is kept.
311 Once these windows are built, they are enumerated to define the events. Hits in other channels
312 are then assigned event numbers if they fall into a given window, again with only the earliest
313 hit being kept. As a result of these steps, the referenced event builder guarantees that events
314 will only contain one hit per `Detector` object. This principle is illustrated in Fig. 6.
315     To implement such a scheme, all hits must be time sorted, a build window must be de-
316 fined, and $n$ channels must be selected as reference channels. Additionally, keeping track of
317 the number of hits that are dropped, we can define an event builder live time for diagnostic

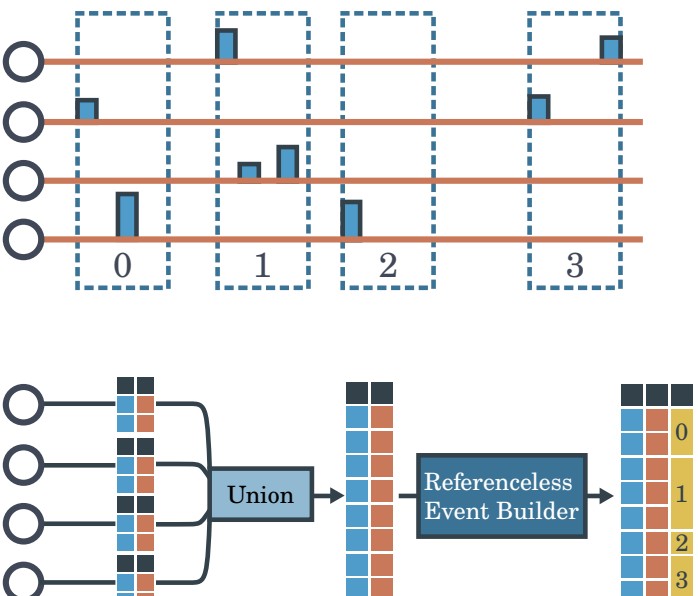

Figure 5: Operation of the referenceless event builder. The top of the figure shows four channels projected along the time axis after they have been digitized. Pulses have a height (light blue) and time (light red), the pulse width corresponds to the sampling rate of the digitizer. The dark blue dashed boxes show the events that would be constructed using the referenceless event builder described in the text. The bottom image shows the list mode data from these same events collected into data frames and then processed in software to achieve the desired result of the top panel.

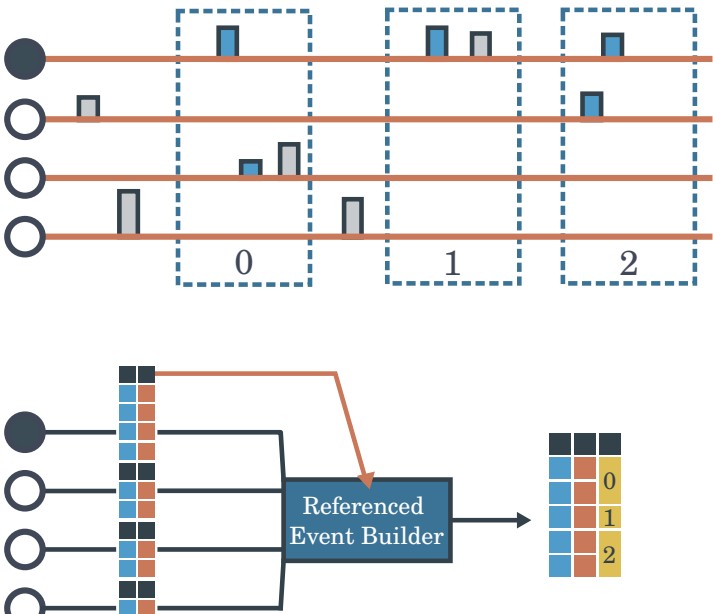

Figure 6: Operation of the referenced event builder. The top of the figure shows four channels (circles) projected along the time axis after they have been digitized. The black circle denotes the reference channel. Pulses have a height (light blue) and time (light red), the pulse width corresponds to the sampling rate of the digitizer. The dark blue dashed boxes show the events that would be constructed using the referenced event builder described in the text. Hits that are dropped due to the referenced event builder are gray. The bottom image shows the list mode data from these same events collected into data frames and then processed in software to achieve the desired result of the top panel.

318   purposes:

$$\text{Live Time} = N_{EB}/N_{Raw}, \tag{4}$$

319   where $N_{EB}$ is the number of hits once the disjoint intervals are built and $N_{Raw}$ is the initial
320   number of hits. If the reference channels are not expected to be in coincidence, then small live
321   times indicate a build window that is too large considering the hit rate. For digital systems,
322   build windows can be made smaller to partially compensated for high dead times by shifting
323   the timestamps of selected channels.

324   Since referenced event building prioritizes specific channels unlike the referenceless case,
325   noisy channels have no impact on coincidences (Sec. 3.1). On the other hand, the deliberate
326   choice to drop hits so that every event has only one hit per considered channel makes it difficult
327   to combine information from multi-channel detectors. In such a case, multiple applications of
328   referenced event building would be required in order to not drop data unnecessarily.

329   Owing to the greater complexity in the implementation of referenced event building, in
330   `sauce` there is a dedicated class for handling its operations.

```
331   # construction
332   eb = sauce.EventBuilder()
333   # either pass a Detector object or array of timestamps.
334   eb.add_timestamps(det0)
335   # construct the build windows.
336   eb.create_build_windows(-500.0, 500.0)
337   # In place modification of the data frame.
338   # Drops all hits after the first and adds event column.
339   eb.assign_events_to_detector_and_drop(det0)
```

340   Although this class performs all of the operations detailing the text, it does not prescribe a spe-
341   cific way to combine the events in separate `Detectors` which will be handled by a dedicated
342   class. So in practice, constructing the disjoint build windows with `eb.create_build_windows`
343   is the last interaction with the event builder a user will have.

344   ## 3.3   Combining Events

345   The operations defined above only enumerate the events for `Detector` objects. A system is
346   still needed to match event numbers between `Detector` objects and combine their respective
347   data frames. On the data frame level, `polars` provides the `join` function (analogous to SQL's
348   `JOIN`), which can be used to combine two data frames on a shared column. Depending on
349   whether the two `Detector` objects should be in coincidence or anti-coincidence, then either
350   an inner-join or anti-join is applied. However, a join takes the columns from each data frame,
351   which all come from list mode data and presumably share column names. This is the issue that
352   was first alluded to in Sec. 2.2, and why the `Detector` class has a required `name` argument. On
353   the inner-join or anti-join, we append this name to all columns in the `Detector`'s data frame.
354   For example, if we want to combine two data frames with names "det0" and "det1", each one
355   will have a "ph" (pulse height) and "ts" (timestamp) column. We keep these columns distinct
356   in the combined frame by renaming them "ph_det0", "ph_det1", "ts_det0", and "ts_det1".

357   The operation to define coincidences is verbose in plain `polars`, so `sauce` abstracts away
358   the details in the `Coincidence` class. First, an `EventBuilder` is constructed, build windows
359   are created, and then it is used to initialize a `Coincidence` object.

```
360   eb = sauce.EventBuilder()
361   eb.add_timestamps(det0)
362   eb.create_build_windows(-500.0, 500.0)
363
364   coin = sauce.Coincidence(eb)
365   # get the coincidences between det0 and det1
366   both = coin[det0, det1]
```

```
367  # get the events in det0 that are anti-coincident with det1
368  anti = coin[det0, ~det1]
```

The compact syntax produces a new `Detector` object containing a data frame with all hits within the individual `Detector` objects that pass the event builder. Anti-coincidences are denoted with ~ before a `Detector` object. All columns associated with `det0` are now named `column_det0` and the same goes for `det1`. Construction of coincidences on demand means that we do not have to waste memory on large, sparse coincidence matrices. It is also at this point that we can finally realize the goal stated in Sec. 2 and compute the timing difference between two detectors in one line of code:

```
376  both["dt"] = both["time_det0"] - both["time_det1"]
```

In fact, any event-by-event processing for the coincidences can be carried out by operations on the coincident `Detector` data frame columns (i.e gates).

## 3.4 Combining Separate Files

Data acquisition is periodically halted to insure data quality and adjust experimental parameters as needed (for example beam retuning). The result is that the total data set will not be one large unbroken file, but will instead be a series of smaller data sets with varying parameters such as integrated beam current. `sauce` relies on this to assume it can load individual runs into memory, but it then becomes necessary to save the run-to-run analysis to build up the final dataset. After a single run has been analyzed and reduced to the relevant counts, `Detector` objects can be tagged and saved to disk for later combination. The tags can then be used to filter the data by run and apply run specific corrections if they are combined via a `detector_union`. For example:

```
389  det_of_interest.tag(run_number, tag_name="run")
390  det_of_interest.save("det_int_run_number.parquet")
391
392  det_of_interest = []
393
394  for run in run_numbers:
395      det = sauce.Detector("det_of_interest").load(f"det_int_{run}.
396      parquet")
397      det_of_interest.append(det)
398
399  det_of_interest = sauce.detector_union("det_of_interest", *
400      det_of_interest)
```

# 4 Example Analysis

Two examples analysis are presented below. Descriptions of the analysis are in this text along with code blocks. The full analysis code can be found in the supplemental materials.

## 4.1 $\alpha$-$\gamma$ Coincidence Measurement

The purpose of this example analysis is to demonstrate the methods of correlating two separate detectors using referenced event building (Sec. 3.2) in order to extract the absolute activity of an $\alpha$ source.

The $\alpha$-decay of $^{241}$Am lends itself to $\alpha$-$\gamma$ coincidence counting, as approximately **35%**, of all decays will coincide with the emission of a **60**-keV $\gamma$-ray from the second excited state to ground state transition of the daughter nucleus $^{237}$Np. Owing to the strong $\gamma$ transition, a modest coincident setup can produce an accurate measurement of the source's activity. Since

412  the coincidence technique, to first order, is not influenced by geometry or detector efficiency,
413  it is a robust and simple precision measurement [17–19].

414        An $\alpha$-$\gamma$ coincidence measurement was carried out to determine the absolute activity of an
415  $^{241}$Am source. The source was purchased from Eckert & Ziegler [20] and consists of $^{241}$Am
416  material electroplated onto a platinum surface in an aluminum A-2 capsule. The quoted NIST
417  traceable activity is **1.230(15)** $\mu$Ci at the **68%** level. The coincidence setup consisted of a
418  silicon surface barrier detector (SSB) located, along with the source, inside of a small vacuum
419  chamber. A CeBr detector is located in atmosphere separated from the vacuum by an acrylic
420  window. A diagram of the setup is shown in Fig. 7. The charge-sensitive preamp signal of the
421  silicon detector and output of the CeBr photomultiplier tube (PMT) were fed directly into a
422  single XIA Pixie-16 module. The list mode data of the Pixie-16 was stored to disk and then
423  read into sauce offline.

424        The goal of this example is to determine the decay rate of the source, which is denoted $N_0$.
425  The observed rate in our detectors ($N$) will be a function of their respective efficiencies ($\epsilon$),
426  decay branching ratios ($B$), and solid angles $\Omega$. For the $\alpha$ and $\gamma$ decay channels we have three
427  equations that can be related to the observed decay rates:

$$N_\alpha = N_0 \Omega_\alpha \sum_{i=1}^{N} B_{\alpha;i} \epsilon_{\alpha;i} \tag{5}$$

$$N_\gamma = N_0 \Omega_\gamma \sum_{i=1}^{N} B_{\gamma;i} \epsilon_{\gamma;i}$$

$$.N_{\alpha\gamma} = N_0 \Omega_\gamma \Omega_\alpha \sum_{i=1}^{N} B_{\gamma;i} B_{\alpha;i} \epsilon_{\alpha\gamma;i}.$$

428        For the $^{241}$Am case, energy discrimination on the **60**-keV $\gamma$-ray and assuming all Branches
429  of $N_\alpha$ can be counted gives:

$$\frac{N_\gamma N_\alpha}{N_{\alpha\gamma}} = N_0. \tag{6}$$

430  It can be seen that source activity is related only to measured quantities (corrected for dead
431  time and background). More details and discussion can be found in Ref. [17].

432        A Parquet[5] file is provided in the supplemental material that can be loaded with sauce.
433  Our goal is to extract $N_\alpha$, $N_\gamma$, and $N_{\alpha\text{-}\gamma}$. Starting with the singles data, $N_\alpha$ is the number of
434  counts observed in the SSB spectrum between the electronic threshold and the highest energy
435  $\alpha$-particle (**5544**-keV in this case). In sauce:

```
run_info = sauce.Run(data_path)
ssb = sauce.Detector("ssb")
# ssb hits are in module 0, channel 0
ssb.find_hits(run_info, module=2, channel=0)
```

440  Since the spectrum is not energy calibrated, we must identify the adc channels by eye to
441  determine the appropriate cut. In a Python repl this can be done using matplotlib, which
442  sauce has a few wrappers in order to call reasonable defaults:

```
import matplotlib.pyplot as plt

x, y = ssb.hist(0, 32000, 32000)
sauce.utils.step(x, y)
plt.show()
# we have now seen where to apply the cut
```

---

[5]https://parquet.apache.org/

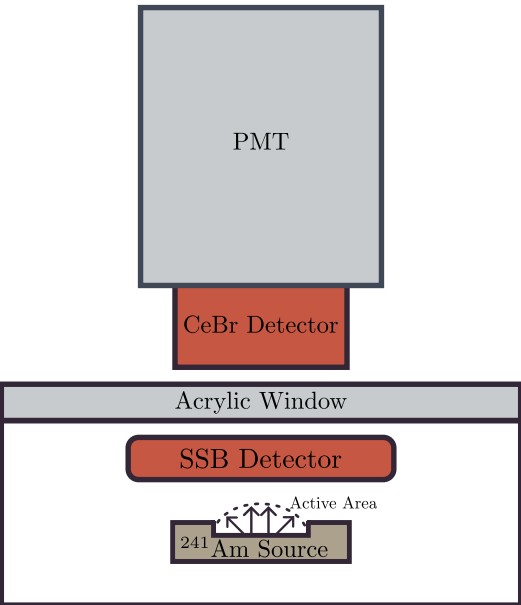

Figure 7: Sketch of the side view of the coincidence setup used to determine the activity of the $^{241}$Am source. A CeBr scintillator detects the $\gamma$-rays coincident with $\alpha$-particles in the silicon surface barrier (SSB) detector. The acrylic window serves as the vacuum interface for chamber. The gap between the the CeBr detector and the acrylic window is only for visualization purposes.

```
449  ssb.apply_cut((0, 3100))
450  # counts are the number of rows
451  c_ssb = ssb.counts()
```

The SSB resolution was limited in this case, and as a result the **5468** keV peak cannot be separated from the **5511** and **5544** keV peaks. Closer inspection of the data prior to the cut shows that pile up was present, but based on just the high energy region is on the level of **0.01%** of all events.

A similar analysis follows for the CeBr detector, but now is limited to the **60** keV region. Deducing $N_\gamma$ for the **60** keV region posed more difficulty as is shown in Fig. 8. The low energy tail comes from interaction of the gamma rays with the source backing, silicon detector, and acrylic window. Depending on the exact energy cut taken, the final deduced activity could fluctuate over **3%**. A relatively narrow region was defined around the maximum of the photo-peak. A side band estimate of the background was taken from the 100 channels adjacent to the **60** keV peak. The code in sauce:

```
463  cebr = sauce.Detector("cebr")
464  cebr.find_hits(run_info, module=2, channel=1)
465
466  # regions of interest
467  peak_region = [1200, 1600]
468  bkg_region = [2000, 2100]
469  peak_bins = peak_region[1] - peak_region[0]
470
471  # copy creates a new detector instance
472  bkg = cebr.copy().apply_cut(bkg_region).counts()
473  bkg_per_bin = bkg / (bkg_region[1] - bkg_region[0])
474
475  cebr.apply_cut(peak_region)
476  c_cebr = cebr.counts() - (peak_bins * bkg_per_bin)
```

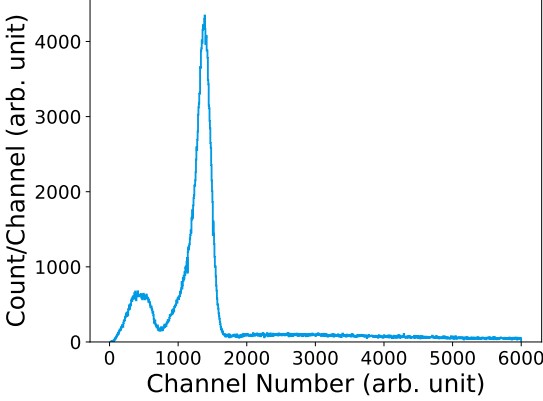

Figure 8: $\gamma$-ray singles spectra of the CeBr detector. Shown are the 33-keV and 60-keV states with heavy low energy tailing induced from the material between the source and the detector.

The "copy" command is necessary to avoid destructive operations on the peak `Detector` object and it copies all information into a fresh "Detector" object.

$N_{\alpha\gamma}$ requires the use of the referenced event builder. In this case, the reference was the energy gated silicon detector. A build window of $\pm 1$ $\mu s$ was selected to account for the $T_{1/2} = 67.2(7)$ [21] [6] half life of the state. The event building dead time (Eq. (4)) was 0.13%. Coincidences were constructed for both for the CeBr peak + tail and peak only energy gates. The random coincident rate was estimated from the flat portion of the spectra. The code:

```
eb = sauce.EventBuilder()
eb.add_timestamps(ssb)
eb.create_build_windows(-1000, 1000)   # nanoseconds

coin = sauce.Coincident(eb)
ssb_cebr = coin[ssb, cebr]
ssb_cebr["dt"] = ssb_cebr["evt_ts_ssb"] - ssb_cebr["evt_ts_cebr"]

peak_region = [-500, 250]
bkg_region = [250, 1000]
peak_bins = peak_region[1] - peak_region[0]

bkg = ssb_cebr.copy().apply_cut(bkg_region, axis="dt").counts()
bkg_per_bin = bkg / (bkg_region[1] - bkg_region[0])

c_coin = ssb_cebr.apply_cut(peak_region, axis="dt") - (peak_bins *
    bkg_per_bin)
```

The last element needed is the total running time of the experiment in seconds. We can closely approximate this by using the timestamps of first and last hits of the run:

```
dt = (run_data.data["evt_ts"][-1] - run_data.data["evt_ts"][0]) / 1e9

activity = ((c_cebr * c_ssb) / c_coin) / dt / 37000
```

Errors from all source (background estimates and counting statistics) were propagated through the calculations via Monte Carlo and can be found in the jupyter notebook. The resulting samples were well described by a normal distribution, and we quote that distributions mean and standard deviation for the reported activities. Our value is $N_0 = 1.284(23)$ $\mu$Ci. The statistical uncertainty is dominated by the counting uncertainty in the number of coincidences. This value is in slight tension with the NIST value of $N_{0;\text{NIST}} = 1.230(15)$ $\mu$Ci. Systematic

513  effects such as non-uniformity of the active area of the $^{241}$Am source were difficult to estimate
514  due to the poorly controlled geometry, but are thought to not amount to more than **5%**.

515       Looking at the code samples for this example, it should be clear that in the case of simple
516  analysis tasks `sauce` leads to a very declarative format. There is not a single level of indention
517  in the whole analysis, meaning we did not have to write loops, conditionals, or dedicated
518  functions even though we essentially started from raw list mode data. Despite not knowing
519  the exact regions to define our gates, it was never necessary to alter any of the code nor update
520  the body of a loop. It serves as a demonstration that the framework outlined in this paper is
521  well matched to its problem domain.

## 4.2  SECAR Focal Plane Detectors

523  The purpose of this example analysis is to demonstrate a more advanced analysis that requires
524  all of the tools presented in this paper. It will require referenced and referenceless event
525  building, tagging, and direct usage of data frame operations. Although it is significantly more
526  complicated then the proceeding example, it shows that `sauce` is not limited to simple use
527  cases.

528       SECAR is a recoil mass separator located at the Facility for Rare Isotope Beams. It is devoted
529  primarily towards measuring $(p, \gamma)$ reactions of astrophysical interest on radioactive nuclei up
530  to $A = 65$ [22]. Through a combination of magnetic dipoles and velocity filters, the intense
531  flux of beam particles are separated from the reaction products. After mass separation, the final
532  section of SECAR consists of two position sensitive micro channel plate (MCP) detectors and
533  stopping detectors including a double sided silicon strip detector (DSSSD) and/or ionization
534  chamber (IC). These combination of detectors are expected to increase the overall rejection of
535  beam particles another 3 orders of magnitude. For the data considered here a hybrid detector
536  (a DSSSD inside of an IC) and two MCPs were used. Descriptions of MCPs similar to those
537  used here can be found in Ref. [23] and likewise for the IC in Ref. [24]. The data is from a
538  **100** $\mu$Ci $^{241}$Am source positioned at the target position of SECAR. The separator was tuned to
539  transmit **4.6** MeV $\alpha$-particles to the final focal plane. The lower energy is a result of amount
540  of material required for such a high activity. A $^{148}$Gd source was also positioned upstream of
541  the hybrid detector to serve as a constant source of counts for gain matching of the DSSSD
542  strips. Due to the low energy of the $\alpha$-particles, the IC was not utilized and no gas was added
543  to the hybrid detector.

544       Our goal is to look at 3 fold coincidences between the two MCPs and the DSSSD. These
545  coincidences require reducing the data by combining the 4 position signals for each MCP, and
546  the 32 strips for the front and back of the DSSSD. The general strategy for making a composite
547  detector in `sauce` is to tag each channel with an identifier, make a union of all of the channels
548  into a single `Detector` object, perform referenceless event building, and then use group by
549  operations on the events to reduce the data in the desired manner.

550       Lets start with the MCPs, since their analysis is relatively simple compared to the DSSSD.
551  For each MCP, the four corners of a resistive anode encoder are read into a channel and are
552  denoted "A", "B", "C", and "D". The relation between the charge detected in the corners and
553  the horizontal $(X)$ and vertical position$(Y)$ is:

$$X \sim \frac{Q_B + Q_C}{Q_A + Q_B + Q_C + Q_D}$$
$$Y \sim \frac{Q_A + Q_B}{Q_A + Q_B + Q_C + Q_D}. \tag{7}$$

---

[6]$T_{1/2}$ was taken from ENSDF (https://www.nndc.bnl.gov/ensdf) which updated the suggested value from
Ref. [21].

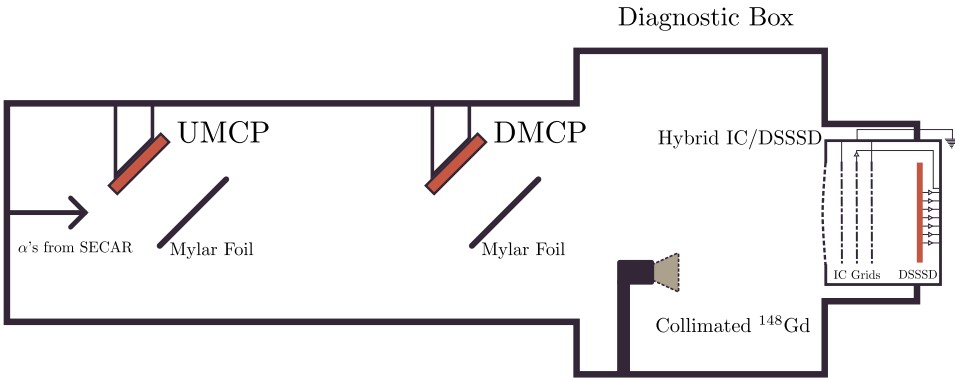

Figure 9: Sketch of the side view of the SECAR focal plane section used in this example. Two position sensitive MCPs are mounted on the top of the chambers to detect electrons produced by the interact of charged particles with aluminized Mylar foils. A $^{148}$Gd source is located in front of the ionization chamber window, but out of the mid-plane of the separator. An additonal Mylar window provides the vacuum interface between the gas filled ionization chamber and the beam line. Note that for the data presented here the IC was evacuated and at vacuum, with only the DSSSD being biased.

These positions must be further calibrated if the position information is to correspond to the $X$ and $Y$ coordinates of the separator; however, this is beyond the scope of the current example. Lets look at the first block of code for the upstream detector:

```
corners = ["A", "B", "C", "D"]
channels = [4, 5, 6, 7]
umcp = [
    sauce.Detector("temp") # this detector will not need a name
    .find_hits(run_data, crate=1, channel=channel, module=2)
    .apply_threshold(2.0)  # cut out noise
    .tag(corner, tag_name="corner")  # apply tag
    for corner, channel in zip(corners, channels)
]
umcp = sauce.detector_union("umcp", umcp)

umcp.build_referenceless_events(500.0)  # 500.0 ns build window
```

The code is terse, but uses a list comprehension to build a list of detectors for each of the corners, applies a small threshold value to get rid of low energy hits, and then tags each detector with the name of the corner. This list is then used to build a single MCP detector via a union operation. Finally referenceless event building is performed with a 500 ns build window. Calculating $X$ and $Y$ from Eq. 7 requires that we only look at events where each of the four channels fired. Complex requirements like this are beyond the operations of `sauce`, and require that we drop down to the data frame level to accomplish our goal. Using `polars` we can do the following: count the number of unique corners in each event and assign that number to a new column, then filter the data frame to keep only the rows that have 4 corners and at least 4 hits, finally if there are multiple hits from a single corner in an event keep the earliest one.

```
581
582   # make a copy of the detector data
583   df = umcp.data
584   # see the polars documentation on windowing functions
585   # this counts the unique corners within the "event_umcp" group,
586   # and assigns them to a new column called "corners"
587   df = df.with_columns(
588       pl.col("corner").n_unique().over(pl.col("event_umcp")).alias("
589     ncorner")
590   )
591   # next drop the rows that don't have each of the four corners
592   df = df.filter((pl.col("ncorner") == 4))
593   # drop multiple hits keeping the earliest
594   df = df.unique(["event_umcp", "corner"], maintain_order=True, keep="
595     first")
596   umcp.data = df # update the detector data frame
```

Positions can be calculated and associated with each event. The function below shows how this is achieved, but there are several equivalent ways to arrive at the same answer. The constants of **0.5** and **8.0** are chosen to give an approximate physical scale, but a means of calibration would be needed to make these positions meaningful.

```
601   def calc_pos(mcp):
602       q_a = get_corner(mcp.data, "A")
603       q_b = get_corner(mcp.data, "B")
604       q_c = get_corner(mcp.data, "C")
605       q_d = get_corner(mcp.data, "D")
606       denom = q_a + q_b + q_c + q_d
607
608       x_raw = ((q_b + q_c) / denom - 0.5) * 8.0
609       y_raw = ((q_a + q_b) / denom - 0.5) * 8.0
610
611       # each event will now have a mean energy and the earliest
612     timestamp
613       mcp.data = mcp.data.group_by("event_" + mcp.name, maintain_order=
614     True).agg(
615           pl.col("adc").mean(),
616           pl.col("evt_ts").min(),
617       )
618
619       # assign the position values.
620       mcp["x_raw"] = x_raw
621       mcp["y_raw"] = y_raw
622
623       return mcp
```

Each side of the DSSSD undergoes a similar procedure to the MCPs, where each strip is tagged and then combined via `detector_union`. It is possible using referenceless event building to correct for inter-strip events, where charge is distributed among adjacent strips [25, 26]. Figure 10 shows these events by plotting the hit that arrives first versus the hit that arrives second. Additional details can be found in supplemental notebook, the correction process is lengthy and requires gain matching the 32 strips per side, combining charge-sharing events, then reducing each column based on its own logic (i.e we keep the strip with the maximum energy in an event, the earliest timestamp, etc.).

Relating the processed MCP and DSSSD hits requires referenced event building. The front strips of the DSSSD are the best candidate for the reference detector. An estimate of the efficiency of the MCPs for the alpha particles can be found by assuming the DSSSD is **100%** efficient and looking at the number of coincidences in the [241]Am peak.

```
636   eb = sauce.EventBuilder()
```

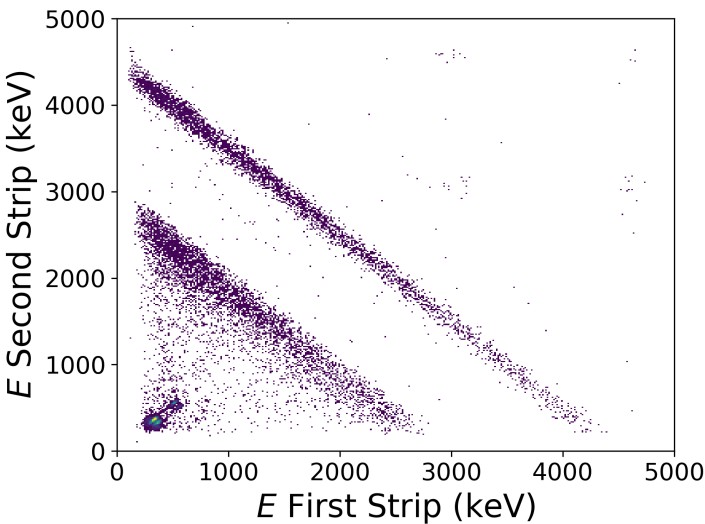

Figure 10: First and second energies for multiplicity equals 2 events from the DSSSD. The charge from inter-strip hits is distributed into two adjacent strips as show here. Strips were gain matched to the $^{241}$Am peak (assumed to be 4600 keV). Low line is from the $^{148}$Gd source (3271 keV before the IC entrance window). Low energy events are assumed to be electronic noise.

```
637  eb.add_timestamps(front)
638  eb.create_build_windows(-500.0, 500.0)
639
640  coin = sauce.Coincidence(eb)
641  umcp_front = coin[umcp, front]
642
643  eff = umcp_front.counts() / front.counts()
```

Doing this we see around **60 − 70%** efficiency for either of the MCPs. It should be clear that at this point we can examine any set of coincidences that we wish, and the data are ready for further, more detailed analysis. Histograms can be saved to a simple text file with:

```
647  x, y = umcp_front.hist(0, 32000, 32000, axis="energy_front")
648  sauce.utils.save_txt_spectrum("dsssd_front_energy.txt", x, y)
```

sauce's role in the analysis is now finished.

## 5   Discussion and Conclusions

Looking over the examples it can been seen that the data pipeline presented throughout this paper has been hand tuned towards low energy nuclear physics experiments. It takes advantage of the relatively low data rates and channel numbers (in the hundreds) to ensure that every portion of the data reduction can be carried out interactively and iteratively. Decisions about how to sort data or combine the information from various detectors can be made while viewing the data, there is no compile and sort cycle typical to many programs in our field, encouraging experimenters to explore the effects of different cuts and event building schemes rather than worry about how to implement these steps. Once the exploratory phase is over, these same commands can easily be amalgamated into scripts and automated. The advantages of such a system are most apparent with trigger less digital DAQs that provide data with minimal structure, but any experiment that lacks a dedicated analysis framework can benefit.

It should be mentioned that similar methods have been described in the literature, see Ref. [27] and references therein, but to the author's knowledge these frameworks only ever attempted to tackle data that had already undergone event building. In this regard, `sauce` is novel. By treating event building dynamically, experimenters gain full control over their analysis. There is no longer a distinction between operations that work event-by-event (gating, timing differences) and ones that can be carried out on a histogram (energy calibration, single channel thresholds).

It may be noted that in the case of the SECAR focal plane analysis there was a significant amount of data frame specific code. Future work could focus on making such operations more compatible with the higher level `Detector` class. Additionally, the author would argue that much of this complexity is coming from the possibility of multiple hits in a channel during one event. This plagued an earlier version of `sauce` that relied solely on referenceless event building. The result was that each analysis would have to specify how to treat multiple hits on a channel-by-channel bases, with some channels needing to keep the most energetic hit, while others would need to keep the earliest. As a result, it was not clear that there was much of an advantage over using an event loop. Referenced event building removed this complexity by aggressively throwing out multiple hits, allowing a more universal analysis framework. Moving forward, it might be beneficial to further diminish the role of referenceless event building and adapt as much analysis as possible to the referenced case.

There is one more potential setback, the techniques described in this paper have been tailored to in memory analysis of list mode data. The focus has been on a fast and responsive data pipeline that can carry out complex analysis interactively, but that is incapable of handling data sets that would exhaust the physical memory of a system. Data sets that are more than 20 GB per run are less common in low energy nuclear physics but by no means rare. Scaling the system described above so that it can handle larger than memory data is well outside the scope of this paper. The core idea of using data frames to provide a flat data structure and complement this structure with purpose built operations, however, can be adapted to use libraries such as `Dask` [28] and `Vaex` [29] that extend data frames to larger than memory data sets. It should be stressed that this step should only be taken as needed, as the in memory methods are typically faster and more flexible. Of all the libraries experimented with during the development of `sauce`, `polars` seemed the best match. It provides excellent speed, most notably for its reading of data from disk, and its data frame interface is well suited with the highly regular structure of list mode data. The out-of-memory solutions were experimented with, but the cost in performance was dramatic. Regardless the point stands: smaller size data sets are a critical part of the low energy nuclear physics mission and tools suited towards their needs have an important place in the field.

# Acknowledgements

The author would like to the C. Brune, C. Iliadis, S. Johnson, R. Longland, and K. Setoodehnia for their reading of this manuscript and helpful comments.

**Funding information** This material is based upon work supported by the U.S. DOE, Office of Science, Office of Nuclear Physics Science, under grants DE-FG02-88ER40387 (Ohio), DE-FG02-97ER41041 (UNC) and DE-FG02-97ER41033 (TUNL). SECAR is supported by the U.S. Department of Energy, Office of Science, Office of Nuclear Physics, under Award Number DE-SC0014384 and by the National Science Foundation under grant No. PHY-1624942 with additional support from PHY 08-22648 (Joint Institute for Nuclear Astrophysics) and PHY-1430152 (JINA-CEE).

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
