# Peer review of "Data Reduction for Low Energy Nuclear Physics Experiments Using Data Frames"

_SciPost Physics Codebases_

## Round 1 · Referee Report · Anonymous (Referee 1) · 2024-8-22

Strengths
1 - dynamic event building is a key feature that improves further analysis steps
2 - enables rapid exploration of analysis strategies
3 - automated analysis with one-line commands is facilitated with this approach
4 - extremely clear exposition of the approach
5 - excellent worked examples
Weaknesses
1 - a graphical user interface would be useful for those researchers who are not proficient in python but who still want to use this novel approach
Report
This work reports on a novel data frame-centered approach to data analysis that includes event building, data exploration, and automated/scripted analysis. The approach is optimized for experiments with low event rates and low channel numbers -- very different than analysis frameworks originating in high energy physics. There is a strong need for such a flexible system, especially in light of the latest low energy nuclear physics accelerator facilities, where rapid data analysis will be key to quick publication of results that is so critical for success these days.
The description of the approach is excellent in that it shows the advantages of its functionalities over existing systems, and it has a very clear exposition of the features. Most exciting is that this enables rapid exploration of different analysis strategies without endless repetition of the compile/sort/display/revise workflow. Additionally, the dynamic event building approach enables full control over all subsequent analysis steps, rather than being forced to use an approach dictated by others. The worked examples are explained is great detail and will be valuable for anyone choosing to use this system in the future.
If this system is well publicized, it could well become a standard at a number of facilities!
Requested changes
no changes requested
Recommendation
Publish (surpasses expectations and criteria for this Journal; among top 10%)
Author: Caleb Marshall on 2024-09-28 [id 4806]
(in reply to Report 2 by Jim Pivarski on 2024-09-10)The author would like to that the referee for their constructive comments which have helped to improve the quality of the manuscript. Points raised by the referee are addressed individually below.
Comment 1
As a minor point, the CDF and D∅ examples in the introduction are dated (end of run in 2011). In the LHC era, high-energy physics event sizes are even larger, but they vary greatly depending on the stage of analysis: https://twiki.cern.ch/twiki/bin/view/CMSPublic/WorkBookDataFormats#EvenT (for CMS) and https://www.epj-conferences.org/articles/epjconf/pdf/2020/21/epjconf_chep2020_06014.pdf (Table 3 for ATLAS). This point in the introduction can be made stronger because the gap in event sizes between low-energy and high-energy nuclear and particle physics is even wider, if we're talking about full events (that high-energy physicists generally don't see).
Reply
Looking at the links provided and comparing to the current generation gamma ray array GRETINA/GRETA it looks like the gap has narrowed, but that is mostly due to the recording of waveform data. The same caveat about full events sizes holds for GRETA as for CMS and Atlas, which makes it difficult to compare the sizes that an experimenter would actually work with. The information added to the manuscript should help the reader put the discussion in a more modern context, and hopefully reinforce the point about fewer channels and low multiplicities. The previous text has been kept since the comparison was more straight forward. The added text reads:
Currently, a stumbling block for this transition to digital DAQs is that low energy nuclear physics frequently adopts the analysis tools and software of the high energy particle physics community, despite the significant structural differences between the data produced by the two fields. As simple example of this difference, consider two of the most prominent experiments from the 1990s in both fields: the CDF and D∅ experiments at Fermilab responsible for the discovery of the top quark in 1995 \cite{top_quark_cdf, top_quark_d0} for the high energy particle physics community, and the construction and operation of Gammasphere at Lawrence Berkeley and Argonne National Labs in the low energy nuclear physics community. Both D∅ and CDF produced events (time correlated collections of all detector signals) of around 200 KB in size \cite{Sinervo_1996}. Gammasphere, however, was producing events of only 100 B \cite{Lee_1990}. In the current era this gap has narrowed with the CMS and ATLAS experiments at the Large Hadron Collider producing events on the order of a MB \cite{cms-events, atlas-events} while the Gamma-Ray Energy Tracking Array (GRETA) has a single crystal event size of 8 kB implying a maximum event size of 960 kB \cite{cromaz-2021}. Despite the narrowing in raw event size, the structure of the events between the two fields remains dramatically different. GRETA's events come from just 120 channels with a majority of an event's size coming from waveform data, while the CMS and ATLAS events contain aggregate information on hundreds of millions of channels. Looking at these examples it is reasonable to say that when comparing the data produced by experiments for low energy nuclear physics and those of high energy particle physics, we are looking at the difference between nearly all information being recorded from hundreds to thousands of channels with low average detector multiplicity versus aggregate information from \textit{hundreds of millions} of channels with high detector multiplicity. The final volume of data produced from these experiments might be comparable, but for low energy nuclear physics these data will constitute hundred of experiments, resulting in average data sets orders of magnitude smaller. Due to these facts, analysis methods tailored to smaller events sizes and fewer channels greatly benefit the field and allow experimenters to utilize computational resources to speed up data exploration and analysis.
Comment 2
Another small point: despite its name, the RDataFrame in CERN's ROOT package isn't a data frame in the sense that the author means. RDataFrame is a functional programming workflow system.
Reply
The text has been updated.
Comment 3
The bigger point that I want to bring up is the back-of-the-envelope calculation in Section 2.1. In an example of 1 hour × 10 kHz triggers of 128 channels × 2 byte resolution,
Moreover, the author says that it "forces us to adopt an event building scheme before even seeing the data," which also isn't necessary. Even though high-energy physics events are pre-built in hardware, this is not a precondition for using jagged array software./
Reply
The text has been altered to clarify that this is a naive use of jagged arrays, and, as pointed out by the referee, a sub-optimal use of the data structure. The original intent was to advocate against the storage format when it is used exactly as described in the text and not against jagged arrays themselves. The author has seen this implementation several times in different labs and those implementations would be classified as jagged arrays, but importantly, and to the referees point, they share almost nothing else in common with the jagged arrays implemented in the Awkward Array package nor do they take advantage of the jaggedness. This implementation is a fixed width array of dynamic arrays (which works out most of the time because there is not actually that much raw data). The assumption made in this case is that the dynamic array takes 3 64 bit numbers to implement (pointer, size, and capacity), making the dataset around 157 GB in the worst case. The references to the referee's work were to point out that this data structure has a name and hopefully gently guide people away from the naive implementation. I have altered the discussion to make the intention more clear, the estimate has also been updated to better reflect the case I was intending to discuss which is not that jagged arrays are an issue, but that clinging to a N-array of channels is.
Comment 4
The "list mode" data collection followed by event-building in software described in this paper can be a good starting point for building jagged arrays in software. Functions that collect contiguous array elements into groups, as described by the event-building procedure, are named "ak.run<sub>lengths</sub>" and "ak.unflatten" in Awkward Array (https://awkward-array.org/doc/main/reference/generated/ak.run_lengths.html and https://awkward-array.org/doc/main/reference/generated/ak.unflatten.html).
Reply
Awkward arrays do present another solution to avoiding large, sparse coincidence matrices. The author has limited experience with the awkward array package (mostly only through Uproot) and that is part of the reason it was not explored more in the manuscript. It is capable software that could be adapted to solve these same problems and could prove to be more succinct for the more convoluted analysis steps (the MCP position processing comes to mind).
Comment 5
The procedure described in the paper uses (slightly) more memory than jagged arrays, after events have been built:
The difference in scaling between the two schemes is that there's one event number per time measurement made by "build<sub>referenceless</sub><sub>events</sub>" in the program listing of Figure 4 (similarly for referenced events), and there is one list offset per distinct event in a jagged array, where events are represented by lists in the jagged array. The two approaches describe the same data:
Reply
I would agree with the point for the referenceless case. For the referenced case the referee is correct that according to Figure 6, there would be a memory overhead from the duplicated event numbers. However, this arises from a figure that shows which detector "hits" are kept accurately, but misleads the reader about the final form of the resulting data frames. Referenced event building only assigns the event numbers and drops duplicates. Later coincident data frames are built using an inner-join between user selected detectors. The resulting data frame has columns for each of the detectors, but there is only one row per event. Obviously there is likely a significant overhead from the data frames themselves, but the resulting data frame, which I would presume has all of the data that the user would want is still the same columnar data structure and would not require further joins or groupbys (at least for the event id).
Comment 6
Jagged arrays impose a memory layout constraint: the data must be time-ordered, but that is asserted by Equation 2 in the paper. The JOIN operation provided by data frame software searches the entire dataset for matching surrogate keys, not taking advantage of the ordered data, which is an O(nlogn) operation. Jagged array operations (instead of JOINs) are O(n) because they take advantage of that constraint. The event-building algorithms described by the paper and in the open-source code do take advantage of the time-order constraint (they look O(n)to me), but Section 3.3 describes uses of Polars's generic JOIN operation.
Reply
The referee is correct, but again due to the misleading image it should be mentioned that the join will only be carried out for the building of the coincident data frame. After this step all operations are carried out on a single dataframe. Looking at Section 4.1, only the line
calls a join (either inner or anti depending on coincidence or anti-coincidence). The desired behavior is to avoid expensive system wide event building, so that these joins and other more intensive manipulations are only called on a small subset of data. However, it is worth exploring how another approach would perform, and if that approach would lead to similarly ergonomic analysis flow.
Comment 7
Finally, there's an ecosystem of tools surrounding Awkward Array that would be directly beneficial here: Akimbo (https://akimbo.readthedocs.io/) provides Awkward Arrays in data frame packages, including Polars, and dask-awkward (https://dask-awkward.readthedocs.io/) provides distributed computing and out-of-core processing, a desideratum mentioned in the paper's last paragraph.
Reply
The performance hit of moving to a distributed framework was too much to justify its inclusion in the original package. I have not experimented with it at great length, mostly because 1) none of the data I have encountered has required it and 2) the software has not been adopted by many people. If the software is adopted by others and the need arises I think the Awkward Array ecosystem could be a quick path to addressing these concerns, but would require a substantial rework to the structure of the package. The paper is both intended to advertise the software and to raise the point that many of us in the community do not need these larger frameworks because modern hardware has caught up to the size of our data sets (excluding experiments that save large amounts of waveform data).
Comment 8
To be clear, I think there's a lot of value in a domain-specific library like sauce, and the current implementation using SQL-like surrogate keys works and makes fairly good use of resources. I think there can be performance gains and better leverage of existing packages if a future version of sauce takes advantage of jagged arrays. The only strong point I want to make is that, currently, the paper describes jagged arrays as a worse solution, for the wrong reasons.
Two minor points regarding software packaging and API:
Reply
The changes to the manuscript will hopefully remove any unintended implication that jagged arrays are poor choice of data structure.
Agreed on both points. The "axis" argument was a holdover from earlier version and only kept to not break existing code, but was a poor choice. This has been switched to "col" and updated throughout the code and manuscript. PyPI does allow distribution names that do not follow the package name, and that would be the preferred route before changing the package name.
Comment 9
One last thing: there are comments in the paper about data files in Supplementary Material. I couldn't find them, and the Jupyter notebooks can't be executed without them.
Reply
It was hoped that these could have been added during the submission process, but that a misunderstanding on the Author's part. A release has been made on github that includes the data files and is now referenced in the manuscript instead of the non-existent supplemental materials.

---

## Round 1 · Referee Report · Jim Pivarski (Referee 2) · 2024-9-10

Strengths
1. Describes novel tools for interactively analyzing low-energy nuclear data in Python/Jupyter.
2. Implements domain-specific tools for event-building in software, allowing the data analysis to be more exploratory because decisions made by a hardware event trigger are irreversible.
3. Well-organized, well-documented Python library.
Weaknesses
1. Points of comparison with high-energy physics misconstrue some things, especially with regard to the use of jagged arrays.
2. The event-building procedure described in this paper, which effectively uses SQL-like surrogate keys, is a valid approach. However, an approach using jagged arrays would also work, likely with better performance, and with the support of an ecosystem that can add distributed and out-of-core computation (a desideratum mentioned in the last paragraph).
Report
For full disclosure, I'm the lead developer of Awkward Array (https://awkward-array.org/), which implements jagged arrays in high-energy physics.
As a minor point, the CDF and D∅ examples in the introduction are dated (end of run in 2011). In the LHC era, high-energy physics event sizes are even larger, but they vary greatly depending on the stage of analysis: https://twiki.cern.ch/twiki/bin/view/CMSPublic/WorkBookDataFormats#EvenT (for CMS) and https://www.epj-conferences.org/articles/epjconf/pdf/2020/21/epjconf_chep2020_06014.pdf (Table 3 for ATLAS). This point in the introduction can be made stronger because the gap in event sizes between low-energy and high-energy nuclear and particle physics is even wider, if we're talking about full events (that high-energy physicists generally don't see).
Another small point: despite its name, the RDataFrame in CERN's ROOT package isn't a data frame in the sense that the author means. RDataFrame is a functional programming workflow system.
The bigger point that I want to bring up is the back-of-the-envelope calculation in Section 2.1. In an example of 1 hour × 10 kHz triggers of 128 channels × 2 byte resolution,
* reading out all channels with every trigger = 10000 × 3600 × 128 × 2 = 9.216 GB ✔
* reading out channels that fired with a 1 byte channel number and 4 byte event number = 10000 × 3600 × (average multiplicity) × (1 + 4 + 2) = 252 MB if the average multiplicity is 1 and 32.256 GB if the average multiplicity is 128 ✔
* the jagged array case is not spelled out, but "assuming 8 byte timestamps" and "would now become 46 GB" likely means 10000 × 3600 × 128 × (8 + 2) = 46.08 GB. That is, the author is assuming that all 128 channels would be read out, with timestamps, in each trigger. Such arrays would not actually be jagged—at least, they wouldn't take advantage of jaggedness.
Moreover, the author says that it "forces us to adopt an event building scheme before even seeing the data," which also isn't necessary. Even though high-energy physics events are pre-built in hardware, this is not a precondition for using jagged array software.
The "list mode" data collection followed by event-building in software described in this paper can be a good starting point for building jagged arrays in software. Functions that collect contiguous array elements into groups, as described by the event-building procedure, are named "ak.run_lengths" and "ak.unflatten" in Awkward Array (https://awkward-array.org/doc/main/reference/generated/ak.run_lengths.html and https://awkward-array.org/doc/main/reference/generated/ak.unflatten.html).
The procedure described in the paper uses (slightly) more memory than jagged arrays, after events have been built:
* assigning 4 byte event numbers to every channel that fired = 10000 × 3600 × (average multiplicity) × (8 + 4 + 2) = 504 MB if the average multiplicity is 1 and 5.04 GB if the average multiplicity is 10
* using 4 byte offsets to represent each event as a list within a jagged array = 10000 × 3600 × ((average multiplicity) × (8 + 2) + 4) = 504 MB if the average multiplicity is 1 and 3.744 GB if the average multiplicity is 10.
The difference in scaling between the two schemes is that there's one event number per time measurement made by "build_referenceless_events" in the program listing of Figure 4 (similarly for referenced events), and there is one list offset per distinct event in a jagged array, where events are represented by lists in the jagged array. The two approaches describe the same data:
* The approach described in the paper is equivalent to database normalization with surrogate keys (the event numbers are a "key" column for an SQL-like table, the data frame), which can be viewed as pointers from nested data to the event that contains them.
* A jagged array approach can be viewed as pointers from each event to the range of nested data it contains. Instead of one pointer per nested datum, it's one pointer per event. This has better scaling if the average multiplicity > 1 (as would be the case for triggered data).
Jagged arrays impose a memory layout constraint: the data must be time-ordered, but that is asserted by Equation 2 in the paper. The JOIN operation provided by data frame software searches the entire dataset for matching surrogate keys, not taking advantage of the ordered data, which is an O(nlogn) operation. Jagged array operations (instead of JOINs) are O(n) because they take advantage of that constraint. The event-building algorithms described by the paper and in the open-source code do take advantage of the time-order constraint (they look O(n) to me), but Section 3.3 describes uses of Polars's generic JOIN operation.
Finally, there's an ecosystem of tools surrounding Awkward Array that would be directly beneficial here: Akimbo (https://akimbo.readthedocs.io/) provides Awkward Arrays in data frame packages, including Polars, and dask-awkward (https://dask-awkward.readthedocs.io/) provides distributed computing and out-of-core processing, a desideratum mentioned in the paper's last paragraph.
To be clear, I think there's a lot of value in a domain-specific library like sauce, and the current implementation using SQL-like surrogate keys works and makes fairly good use of resources. I think there can be performance gains and better leverage of existing packages if a future version of sauce takes advantage of jagged arrays. The only strong point I want to make is that, currently, the paper describes jagged arrays as a worse solution, for the wrong reasons.
Two minor points regarding software packaging and API:
* The name "sauce" is already used in PyPI by an active project: https://pypi.org/project/sauce/#history Unless the package described by this paper changes its name, it will never be possible to install it with pip, which most Python users expect to do. Installing from GitHub is not a user-friendly option.
* The use of "axis" as a function argument to mean "column" goes against conventions established by NumPy, Pandas, and others. It would be confusing.
One last thing: there are comments in the paper about data files in Supplementary Material. I couldn't find them, and the Jupyter notebooks can't be executed without them.
Requested changes
1. Improve the introduction using LHC event sizes.
2. Remove the sentence about CERN ROOT's RDataFrame being a data frame. ROOT is good software, but it doesn't have functionality that is similar to what is being described here, which is array-oriented programming on data frames.
3. Remove or somehow rework the text about jagged arrays, which describes them as a severe disadvantage. Whether or not you add text describing how they can be used in a future version of sauce is up to you: it depends on how much you want to investigate before publishing this paper.
Other suggested changes—investigating the use of Awkward Array, Akimbo, and dask-awkward, renaming sauce, and renaming the "axis" function argument—are about the software, not the paper.
Recommendation
Ask for minor revision

---

## Editorial Decision

unknown